# Effects of High Voltage Electrical Discharge (HVED) on Endogenous Hormone and Polyphenol Profile in Wheat

**DOI:** 10.3390/plants12061235

**Published:** 2023-03-08

**Authors:** Tihana Marček, Kamirán Áron Hamow, Tibor Janda, Eva Darko

**Affiliations:** 1Faculty of Food Technology, Josip Juraj Strossmayer University of Osijek, Franje Kuhača 18, 31000 Osijek, Croatia; 2Agricultural Institute, Centre for Agricultural Research, ELKH, 2462 Martonvásár, Hungary

**Keywords:** high voltage electrical discharge (HVED), germination, hormones, polyphenols, root, shoot, wheat

## Abstract

High voltage electrical discharge (HVED) is an eco-friendly low-cost method based on the creation of plasma-activated water (PAW) through the release of electrical discharge in water which results in the formation of reactive particles. Recent studies have reported that such novel plasma technologies promote germination and growth but their hormonal and metabolic background is still not known. In the present work, the HVED-induced hormonal and metabolic changes were studied during the germination of wheat seedlings. Hormonal changes including abscisic acid (ABA), gibberellic acids (GAs), indol acetic acid (IAA) and jasmonic acid (JA) and the polyphenol responses were detected in the early (2nd day) and late (5th day) germination phases of wheat as well as their redistribution in shoot and root. HVED treatment significantly stimulated germination and growth both in the shoot and root. The root early response to HVED involved the upregulation of ABA and increased phaseic and ferulic acid content, while the active form of gibberellic acid (GA1) was downregulated. In the later phase (5th day of germination), HVED had a stimulatory effect on the production of benzoic and salicylic acid. The shoot showed a different response: HVED induced the synthesis of JA_Le_Ile, an active form of JA, and provoked the biosynthesis of cinnamic, *p*-coumaric and caffeic acid in both phases of germination. Surprisingly, in 2-day-old shoots, HVED decreased the GA20 levels, being intermediate in the synthesis of bioactive gibberellins. These HVED-provoked metabolic changes indicated a stress-related response that could contribute to germination in wheat.

## 1. Introduction

The major idea of agricultural sustainability is to fuse ecological and agronomic practice, farmer skills and knowledge to reduce the utilization of pesticides and fertilizers in the food production chain [1]. Today, in agricultural sustainability programs, low-cost beneficial innovative solutions with a minimum impact on the environment are receiving more attention. Numerous reports highlight an improvement in seed invigoration after the application of physical methods such as electromagnetic technologies (EMFs) and magnetic field technologies (MFs) [2,3], ionizing radiation (IR) [4,5] and various cold plasma technologies (CPTs) [6,7,8].

High voltage electrical discharge (HVED) is a physical, non-thermal, hydropriming technique that causes modification in the physicochemical structure of water during electrical discharge generation exposure [9]. An altered water structure is a result of ionization and dissociation reactions upon which reactive oxygen species (ROS) and reactive nitrogen species (RNS) may be created forming plasma-activated water (PAW) [10]. The chemical content of PAW is not always equal and the concentration of long-lifetime particles such as H_2_O_2_, NO_2_^−^ and NO_3_^−^ may depend on the experimental conditions of PAW preparation [11]. In addition, as observed via SEM micrographs, the HVED treatment induced ruptures on the seed coat [12]. These two events could be responsible for seed dormancy disruption and better germination properties [13,14,15]. Accordingly, the HVED treatment promoted the germination and growth of wheat seedlings, as we observed previously [12]. Furthermore, this effect could also be observed even under drought (15, 20 and 30% PEG) and salt (90, 160 and 230 mM NaCl) stress conditions, although the final success of HVED depended on the stress intensity [12]. These results indicated that the HVED affected and was connected to the stress response of plants.

Seed germination is a particularly important stage in plant development which is characterized by fast water uptake into dry seeds which activates several physiological and metabolic processes, resulting in radicle and bud protrusion [16]. Each event of germination is precisely regulated with multiple types of phytohormones such as auxin (IAA), abscisic acid (ABA), gibberellins (GAs), jasmonates (JAs) and salicylic acid (SA) [17]. GA and ABA are major seed dormancy and germination regulators in optimal and stressful environments. In the early stage of germination, seed hydration triggers gibberellin biosynthesis in embryo meristems, which is subsequently directed into the aleuronic layer and endosperm, whereas it induces the synthesis of α-amylase, a major hydrolyzation enzyme, leading to the disruption of seed coat dormancy [18]. GA induces the expansion of embryo cells, radicle protrusion and the elongation of young cells, and may also increase seed osmotic and water potential [19,20]. GAs synthesize through different intermediaries including GA53 and G12, which convert to GA9 and GA20. Finally, through 3-β-hydroxylation, GA1 and GA4 active compounds are generated [21,22]. Cui et al. [23] found that 1 min of exposure to atmospheric pressure cold plasma was enough to stimulate *Arabidopsis* growth and the production of GA.

ABA affects germination through the suppression of the events controlled by gibberellins. Phaseic acid (PA) and dihydrophaseic acid (*d*HPA) are compounds of ABA catabolism generated through the oxidation of 8′-hydroxy ABA which is the primary route of ABA decomposition [24]. Attri et al. [25] reported that non-thermal plasma treatment (NTP) caused a shift in the GA/ABA ratio which was reflected in the promotion of the germination of radish seeds. In heat-exposed rice seeds, NTP induced germination by suppressing ABA anabolic and inducing ABA catabolic gene expression [26]. One of the important regulators of seed dormancy is auxin (indole-3-acidic acid, IAA). Many studies have revealed a crosstalk between auxin and ABA signaling in keeping the dormancy [27,28,29,30]. The relationship between cell elongation, IAA content and plasma treatment has been observed in *Arabidopsis* [31]. The authors found that the active oxygen particles derived from air plasma promoted the expression of EXPA8 and EXPA20 expansin genes important in acid-induced growth [32]. Jasmonic acid, jasmonate isoleucine conjugate (JA-Ile) and methyl jasmonate (MeJA) belong to jasmonate (JA) pathways that originate from α-linolenic acid esterified in the galactolipids of thylakoids [33]. JA_Ile is formed by the conjugation of JA and isoleucine [34]. JA often operates in conjunction with other phytohormones and participates in complex regulatory network signaling pathways [35,36]. PAW watered tomato plants showed higher activity of 2-oxophytodienoic acid reductase (OPR1), phenylalanine ammonia-lyase (PAL) and allene oxidase synthase (AOS), key enzymes in SA and JA synthesis [37].

Phenylpropanoid metabolism plays a major role in the synthesis of a large number of compounds important for plant growth, development and stress response. Phenolic acids are separated into hydroxybenzoic and hydroxycinnamic acids [38]. Salicylic acid (SA, 2-hydroxy-BA) belongs to hydroxybenzoic acids and it is a plant growth regulator involved in the plant immunity response [39]. SA can be synthesized either from shikimate, converted into chorismate and isochorismate, or from *p*-coumaric acid (*p*-CA) via the phenylpropanoid pathway [38,40]. Phenylalanine (Phe), originating from the shikimate pathway, is a starting point in phenylpropanoid metabolism. The deamination of Phe creates cinnamic acid which via hydroxylation is converted into *p*-coumaric acid (*p*-CA) [40]. Ferulic acid (FA), caffeic acid (CFA) and *p*-CA are esters of hydroxycinnamic acids (C6-C3) naturally present in cell walls [40,41]. Jasim et al. [42] found that the application of DBD plasma on basil seeds promoted the expression of genes important in the phenylpropanoid pathway (4-coumarate coA ligase, 4CL, cinnamate 4-hydroxylase, C4H, and charicol o-methyl transferase, CVOMT), which indicated that the compounds of phenylpropanoid pathways may also play an important role in germination.

As indicated above, the HVED treatment stimulated the germination and growth of wheat under both control and stress conditions [12]. To better understand the mode of action of HVED treatment on germination and seedling development, the aim of this work was to follow the alterations in hormone profiles (ABA, GAs, IAA, JA) and their derivates, phenylpropanoids and benzoic acids in wheat. The effect was studied in a time-dependent manner, on the 2nd and 5th day of germination. Moreover, we wanted to compare the effect of HVED treatment on the spatial distribution of phytohormones and their substituents and to check whether HVED affects specific organ (root and shoot) responses. Since the evidence of the HVED impact on hormonal response and secondary metabolism has not been presented yet, the results of this study could contribute to understanding the role of HVED in hormonal balance in young wheat seedlings.

## 2. Results

### 2.1. Effect of HVED Treatment on Germination and Growth

The HVED treatment promoted germination and growth in a time-dependent manner, as indicated by the results of statistical analysis (Appendix A). The most notable changes in germination percentage were recorded on the 1st day in the HVED group in which germination was 28.4% higher than in the control (Figure 1). Although the differences in the germination percentage between the control and HVED groups became less pronounced with time, the germination % remained higher in HVED seeds as compared to the control seeds during the 5-day experimental period (Figure 1). Shoot and root growth were also improved in the HVED group (Figure 2A,B). Interestingly, the bigger differences in growth dynamics between the control and HVED group were observed within the 2–4 days, while differences in growth diminished on the 5th day. For instance, after the 2nd day, the HVED shoots were 31.2% higher than the control, while on the 5th day, the difference was only 2%. In the root, 2-day-old HVED roots were 10.6% longer than the untreated HVED, whilst after 5 days the HVED group had 6% longer roots compared to the control.

### 2.2. HVED Impact on Hormonal and Metabolite Profile

In the shoot, treatment had a significant impact on most of the phenylpropanoid substrates (except FA), JA and its conjugates and BA, as indicated by analysis of variance (Appendix A). On the other hand, time affected all variables except for *p*-CA acid, indicating that its reservoirs in shoots are time-independent. Finally, treatment and time interactions were noticed only for CA and BA, suggesting interconnections in the β-oxidation pathway of the BA metabolic route. In the root, treatment affected ABA, PA, *d*HPA, FA, GA1, BA and *p*HBA content, while variations in the time-dependent response were found for most of the compounds with the exception of *p*-CA, IAA and *p*HBA amount. The results of the statistical analysis are presented in the Appendix A. The significant effects for treatment and time points were observed for ABA, PA, FA, GA1, 3,4-*d*HBA and BA, respectively. Appendix A present the numerical values of hormones and polyphenols in shoots and roots.

The accumulation of ABA and its catabolites in the shoot and root was higher in the earlier time point (2nd day) than on the 5th day in the control and HVED groups (Figure 3A,B). In addition, ABA metabolism differed in the root of the control and HVED groups. Namely, HVED treatment stimulated higher ABA and PA content production in 2-day-old roots compared to the control, while later (5th day) there were no differences between treatments (Figure 3B). The level of *d*HPA was higher under the control than after HVED application, in both the root (2nd and 5th day) and shoot (2nd day), referring to slower ABA decomposition under HVED. The GA distribution was tissue-specific and time-dependent as indicated via analysis of variance (Appendix A). In the shoot, bioactive GAs (GA4, GA3 and GA1) and their intermediates (GA20 and GA8) were detected except for GA4 which remained undetected on the 5th day of growth (Figure 3C). Similar to ABA metabolism, the production of all GA derivates in the shoot was higher on the 2nd day compared to the 5th day in both treatment groups. Unlike the shoot, the root contained only a few compounds (GA1 and GA8) involved in the early-13-hydroxylation pathway. The amount of GA1 was higher on the 2nd than on the 5th day in the control and HVED group, whilst its inactive form GA8 was detected only on the 2nd day (Figure 3D). Comparing GA accumulation within the one-time point, the 2-day-old control plants of shoots and roots showed a higher concentration of GA20 and GA1 than the HVED-treated plants (Figure 3C,D).

The biosynthesis of JA and JA_Le_Ile decreased over time in shoot and root, respectively (Figure 3E,F). Five-day-old seedlings had a lower amount of JA and JA-Le_Ile than two-day-old seedlings under both treatment groups (control and HVED). The HVED treatment triggered the overproduction of JA and JA_Ile derivates only in the shoot. Namely, the percentage of JA and JA_ILE derivates after the 2nd day was 23.5% and 68%, higher in the HVED group than in the control group, while later (5th day) their concentration increased by 66% and 174%, respectively. The control and HVED-exposed 2-day-old shoots contained higher IAA content than the 5-day-old shoots (Figure 3G), indicating IAA decomposition during the time. For the root, significant differences in IAA synthesis were not detected between days of growth or between treatments (Figure 3H).

The phenylpropanoid distribution in the root and shoot was more time-dependent than treatment-dependent (Figure 4A,B). The amount of CA was higher on the 2nd day than the 5th day of growth in both the root and shoot, independent of the treatment. Similarly to CA, the FA amounts were higher on the 2nd day in the root while an inverse response in FA amounts was obtained for the shoot; FA values were lower on the 2nd day compared to the 5th day’s values. HVED treatment provided a significant increase in CA, *p*-CA and CFA in the shoot and FA in the root on the 2nd day of growth. The differences disappeared on the 5th day, but became notable for FA whereby the amount was higher in the HVED group.

The results revealed a time-dependent accumulation of several BA compounds without treatment influence in the shoots (Figure 4C). In the shoot, the amount of 3,4-dHBA was higher on the 2nd day as compared to those data measured on the 5th days, independent of treatment, while inverse tendencies were observed for SA, *p*HBA and 2,6-dHBA (their amount was higher on the 5th day). The early effect of HVED on the shoot’s polyphenol response was notable only for BA whose levels increased compared to the control on the 2nd day. Later, HVED had an inhibitory effect on BA biosynthesis in shoots that showed a lower BA amount than the control. In roots, the HVED treatment immediately (2nd day) triggered the higher accumulation of *p*HBA but negatively affected 3,4-*d*HBA amounts in comparison to the control (Figure 4D). Unlike the control, the HVED group had a higher amount of SA and BA on the 5th day. Moreover, the same trend was detected during the time where the amount of BA and SA remained significantly higher in HVED-treated roots.

### 2.3. Comparative Statistical Analyses

To reveal the connection between the effect of treatment during the times of germination (first three days), shoot and root growth, a comparative statistical analysis was performed (Table 1). The treatment was positively correlated with germination (G1-G3), shoot (S2-S5) and root (R2-R5) length during the time. A strong positive association was also notable for germination and growth. Comparative statistical analysis was also used to verify the interactions of treatments on growth parameters and changes in hormone and metabolite amounts in the time frame for shoot and root, separately (Table 2 and Table 3). In 2-year-old shoots, a correlation was found between treatments and the phenylpropanoids (except *p*-CA), BA (except *p*HBA and 3,4-*d*HBA), JA, germination percentage and shoot length (Table 2). During the time (5th day), treatments showed an association with the accumulation of JA and its conjugates, GA1, germination and shoot length, but a negative correlation with BA and 3,4 *d*HBA amount. On the 2nd day, a positive association was found between germination and shoot length (0.83 ***), CA (0.71 *), BA (0.75 *), *p*-CA (0.70 *) and SA (0.82 *) (Table 2). Moreover, shoot length was connected with CA (0.85 **), BA (0.91 **), *p*-CA (0.83 **), CFA (0.69 *), FA (0.95 ***) and JA (0.92 **), indicating improved activation of the shikimate acid pathway at an early time point. Most associations connected with shoot early metabolic changes were found for CA and BA. Thus, CA correlated with benzoic acid (0.95 ***), *p*-CA (0.70 *), CFA (0.64 *), FA (0.89 *), JA (0.99 ***) and JA_Le_Ile (0.73 *), respectively. A positive association was noticed for BA and *p*-CA (0.86 *), FA (0.94 **), JA (0.93 **) and the active form of JA (0.71 *), respectively. Over time, the number and nature of the relationship between metabolites and hormone amount were lower. Namely, on the 5th day, a negative association was noticed between *p*-CA and 2,6-*d*HBA (−0.76 *) and between BA and GA1 (−0.83 *). Moreover, a negative association was also noticed for FA and BA (−0.94 **) and between FA and GA20 (−0.89 *), respectively. A strong positive connection (*p* ≤ 0.001) was seen between BA and 3,4-*d*HBA, CFA and SA, FA and GA1, JA and JA_Le_Ile and finally between JA and PA.

Table 3 presents the correlation coefficients for root-related metabolites and hormones. In 2-day-old roots, a correlation existed between treatments and the amount of *p*HBA, FA and PA, ABA, germination percentage and root length, while the production of GA1 and 3,4-*d*HBA was downregulated by treatment. Later (5th day), a positive association was found in the amount of CA, BA, 2,6-*d*HBA, SA, GA1 and germination percentage, while the treatment was negatively correlated with *d*HPA. Correlation analyses revealed a relationship between germination and ABA content (0.99 ***) in the early stage of growth (2nd day), *p*HBA amount (0.78 *) and PA (0.71 *), while germination had an opposite effect on 3,4-*d*HBA synthesis (−0.74 *). ABA, FA, PA and *p*HBA amounts were connected with root length. The amount of IAA was correlated with FA (0.83*) and was negatively correlated with JA (−0.86 *) and *d*HPA (−0.89 *) suggesting a greater activation of metabolic routes connected with seed development. PA showed a negative correlation with 3,4-*d*HBA (−0.92 *) but a positive correlation with *p*HBA (0.72 *), FA (0.83 *) and ABA (0.83 *). Moreover, in 2-day-old roots, the *d*HBA amount correlated with JA content (0.90 *) and GA1 (0.99 ***), but it had the opposite relationship with *p*-CA amount (−0.90 *), IAA (−0.89 *) and FA (−0.98 **), respectively. On the other hand, JA levels were correlated with GA1 (0.90 *) and GA8 (0.86 *) but were negatively connected with IAA, *p*-CA and FA, indicating the crosstalk between JA signaling and gibberellin metabolism in the root. A positive linkage (0.88 *) was evident for CA and CFA while the amount of *p*HBA was in a negative association with 3,4-*d*HBA (−0.78 *) suggesting that the hydroxylation of benzoic acids depends on the availability of precursor molecules. Finally, the FA amount was negatively connected with GA1, JA and *d*HPA but positively connected with ABA and PA, indicating an early ABA regulatory role in polyphenol root induction. The germination percentage in 5-day-old roots correlated with GA1 growth but showed a negative connection with the *d*HPA level. The greatest number of associations was noticed for CA and BA in the root on the 5th day. Thus, CA was positively correlated with BA (0.78 *), SA (0.68 *) and *p*-CA (0.72 *) and was negatively correlated with *d*HPA (−0.89 **). Furthermore, BA levels were associated with SA (0.74 *) and 2,6-*d*HBA (0.76 *) while a negative connection was detected with *d*HBA (−0.81 *). The other positive connections (*p* ≤ 0.01) were noticed between IAA and 3,4-*d*HBA, *p*-HBA and *p*-CA, SA and 2,6-*d*HBA amount, root length and 3,4-*d*HBA, respectively. In contrast, negative associations were observed for *d*HPA and GA1 (−0.79 *), 2,6-*d*HBA (−0.71 *), CA (−0.89 **), BA (−0.81 **) and finally between JA and ABA amount (−0.83 *).

### 2.4. PCA Analyses

Figure 5A shows the percentage of the total variance and principal components (PCs) and their interactions with hormones and metabolites under control and HVED treatment for roots during the time. As the PCA analyses did not detect the explanatory variables for the shoot, the factor loadings are presented only in Appendix A. For the root, the variable projection is presented for the first two components, contributing 93.19% of the total variance (Figure 5A). The cumulative contribution to the total variation in PC1 was 79.34% (Appendix A). The PC1 was characterized by positive loadings of CA and BA, 3,4-*d*HBA, *p*-CA, *d*HPA, JA and its conjugates, ABA and IAA, respectively. Negative associations were evident for *p*HBA, SA, 2,6-*d*HBA, CF and FA, GA3, GA20 and root length. The PC2 was explained by negative loadings of 3,4-*d*HBA and positive loadings of PA and germination (2nd day), while the positive loading was obtained in PC3 for GA8. The projection of the corresponding plots reveals four clusters separated based on treatment and days of sampling (Figure 5B). The first cluster (A) belongs to the 2-day-old HVED roots (2d-H) segregated based on the higher amount of ABA, PA, FA and *p*HBA but the lower accumulation of *d*HPA and 3,4-*d*HBA compared to the 2-day-old control group (2d-0) (cluster B). The separation of 5-day-old roots into clusters C (5d-0) and D (5d-H) was the result of HVED treatment on root length, *d*HPA, SA and BA amount. Namely, the roots of HVED-treated seeds (cluster D) were longer and showed increased SA and BA levels and decreased *d*HPA compared to the control (cluster C). HVED-treated groups (A, D) were distinguished through specific time-frame responses. In group A, HVED promoted increased ABA, *p*HBA, *d*HBA, FA and PA accumulation, while group D had higher SA and BA concentrations under HVED. Finally, clusters B and C were apart due to a difference in the production of ABA and its catabolites, root length, GA8, JA and its derivates, CA, CFA and FA, whose production was upregulated earlier.

## 3. Discussion

### 3.1. HVED Impact on Germination and Growth

In the present study, the metabolic background of the HVED-induced stimulation in germination and growth was studied. The stimulatory effect of HVED on seed germination was manifested in the shortening of the imbibition time, improving germination and root and shoot growth within the early (2–5 days) stages of growth (Figure 1 and Figure 2). Although the HVED-induced treatment improved seed germination, as has already been observed in wheat [12,43,44], *Nasturtium seeds* [45] and oilseed rape [46], the physiological and metabolic background have not been revealed yet. The stimulatory effect of HVED is based on the fact that this treatment weakens the seed surface layers, thus promoting seed invigoration and growth [47]. The exposure of seeds to HVED causes seed surface modification which shortens the imbibition time and increases wettability leading to faster oxygen diffusion into the embryo [15]. HVED treatment creates plasma-activated water (PAW) rich in ROS and RNS particles which can also affect germination [48]. Moreover, the PAW creates a higher amount of hydrogen peroxide (H_2_O_2_) and nitric oxides (NO), longer half-life molecules which may alter the phytohormone balance in the seeds and promote germination [49]. Recently, Mildaziene and Sera [50] and Priatama et al. [51] observed that the accumulation of ROS and RNS triggers changes in proteomic, metabolomic and transcriptomic levels and also affects phytohormone levels. The cavitation bubbles which occur during the HVED treatment create UV emission which could also be meritorious for germination dormancy disruption [9,44]. However, it should also be mentioned that the success of germination depended on the number of shots. Ji et al. [52] found that one or five energy shots had a beneficial effect on germination, while seeds treated with ten shots decreased growth and germination. These observations were found in spinach. In the present study, the energy input was 30 kV, suggesting that such a huge energy input still improved the germination of wheat seeds. In addition, HVED treatment induced several changes at the metabolic level too.

Correlation analysis reveals a positive association between germination and *p*-CA, CA, GA1, BA and SA in the shoot (2nd and 5th day) (Table 2). The positive crosstalk between GA and SA interaction in germination and early growth has been published for *Arabidopsis thaliana* [53]. Moreover, phenol compounds (SA, BA, *p*-CA and CA) were also important inducers of germination. Since all of these compounds originate from the Phe metabolic route, our results confirm the connection with complex hormone interaction. Curiously, germination in 2-day-old roots positively correlated with ABA, *p*-HBA and PA, while the 5-day-old root germination percentage was associated with GA1 indicating different regulation pathways of germination during the time (Table 3). These results support the observations of Pei et al. [54], who found that HVED treatment generates ROS and H_2_O_2_ which participate in ABA signaling. Moreover, it is also possible that the influence of H_2_O_2_ diminishes over time, which might be a reason for a decreased ABA amount in 5-day-old roots (Figure 3B). Correlation analysis displays a positive linkage between JA and shoot length (2nd day), indicating that growth promotion could be ascribed to the early activation of the JA regulation pathway. Over time, GA1 also contributed to shoot growth (*p <* 0.05), suggesting that gibberellins act through jasmonoyl-isoleucine in the coordination of growth. In the root, early growth was promoted by ABA, PA, FA and *p*HBA while, over time (5th day), length was associated with 3,4-*d*HBA (Table 3). This indicates that the ABA pathway and phenolic compounds have a similar action on early root formation. Moreover, a linkage between ABA and root growth could be also connected with ABA growth regulation under dark conditions since the seeds were kept for 48 h in the darkness before sampling. The ABA-induced growth occurs in the process of skotomorphogenesis where the cells of hypocotyl rapidly elongate [55]. In both 5-day-old shoot and root germination, the percentage was positively regulated by the GA1 amount (Table 2 and Table 3). According to Vishal and Kumar [56], the presence of GA-coding genes in the growing tissues suggests that active forms of GA can be produced in situ.

According to the results, different metabolic responses were found in the root and shoot. Thus, they are discussed here separately.

### 3.2. Root Hormonal and Polyphenol Response

The early root response revealed the antagonistic interaction in ABA/GA signaling in both the control and HVED groups, with slight alterations in the rate of ABA decomposition and GA1 formation. Namely, on the 2nd day, HVED increased ABA (*p* < 0.05, r = 0.88) and PA (*p* < 0.01, r = 0.87) content but inhibited the biosynthesis of the active form of gibberellins (GA1) (*p* < 0.05, r = −0.88) (Figure 3B,D and Table 3), suggesting that the ABA hormone suppresses GA biogenesis. Jacobsen et al. [57] stated that a low concentration of GAs could be the result of high ABA amounts due to changes in the conversion of their common precursor geranylgeranyl diphosphate (GGDP) to ent-kaurene. In contrast, the control displayed a higher GA1 level which might be connected with earlier activation of the 13-hydroxylation gibberellin pathway. In general, a lower ABA/GA ratio under optimal conditions participates in seed dormancy breakdown events [58]. In this study, the ABA/GA1 ratio in HVED roots was higher (1.33) compared to the control (0.86), indicating that HVED modifies the ABA pathway at the expense of GA routes. As the opposite, the GA3 concentration in spinach increased after non-thermal (NTP) treatment [52]. In another study, NTP application increased the expression of the LEA1, SnRK2 and P5CS drought-responsible genes and higher ABA generation in wheat seedlings [59]. Correlation analysis also showed a positive linkage of germination with ABA (*p* < 0.001, r = 0.99) and PA (*p* < 0.05, r = 0.71) (Table 3). A higher PA content and higher germination percentage on the 2nd day suggests that HVED in the root triggers the dormancy-breaking events compared to the control group earlier in which the ABA degradation was complete due to its final catabolite *d*HPA. As provided by PCA analysis, the 2-day-old control group (B cluster) was separated from the HVED group (A cluster) based on a higher *d*HPA amount (Figure 5A,B). Ferulic acid (FA), a phenylpropanoid substrate of the β-oxidative pathway, creates cross-connections between sugars, polymers and the lignin of the cell wall, increasing the mechanical strength especially important in the conditions of pathogen attack [60]. Our results showed a higher FA amount in 2-day-old-HVED roots which might be connected with an early root response to ROS and RNS in PAW medium and the stress response (Figure 4B). In the study of Cortese et al. [61], PAW caused an increase in cytosolic Ca^2+^ concentration in *Arabidopsis*. The role of Ca^2+^ signaling in the perception of ROS and RNS was described in [62,63]. In this context, it seems that HVED takes part in Ca^2+^-dependent signaling transduction. However, further investigations in this direction are required. Correlation analyses display the positive connections between FA content and root growth, and ABA and FA amount, respectively (Table 3). Yukiko et al. [64] and Marchiosi et al. [65] reported a dual role in growth regulation between hormones and polyphenols, indicating that some regulatory polyphenols inhibit growth by binding GAs, while others, through ABA signaling, promote it. Considering this, it seems that in early root growth, FA biosynthesis is ABA-dependent.

In 5-day-old roots, HVED promoted the accumulation of BA and SA compared to untreated roots. The separation of these two groups was presented via PCA analysis (Figure 5A,B). Correlation analysis also showed a positive linkage between BA and SA (*p* < 0.05, r = 0.74), suggesting that HVED for SA synthesis activates the phenylpropanoid rather than the chorismate pathway [66]. In wheat, plasma treatment increased the activity of Phe ammonia lyase (PAL), which catalyzes the deamination of Phe to CA [43,67]. Additionally, in tomatoes, PAW induced the tissue-specific expression of genes involved in JA and SA synthesis [37]. In the same study, shoots showed an induction of 12-oxophytodienoic acid reductase (opr 1) and allene oxidase synthase (aos) JA regulatory genes, while in PAW-treated roots their expression was suppressed. In contrast, the expression of phenylalanine ammonia-lyase (pal), the SA encoding gene, in PAW-treated root and shoot was higher. However, such a conclusion should be interpreted with caution. According to Widhalm and Dudareva [39], the generation of SA might be the result of the utilization of crosstalk intermediate compounds in chorismate and Phe pathways via feedback inhibition.

Correlation analysis reveals the difference between 2-day-old and 5-day-old roots in JA regulation and the type of metabolite expression. At the first time point, JA was positively regulated by *d*HPA (*p* < 0.05, r = 0.90), GA1 (*p* < 0.05, r = 0.90) and GA8 (*p* < 0.05, r = 0.86), while later (5th day) a negative correlation (*p* < 0.05, r = −0.83) existed between JA and ABA (Table 3). This leads to the conclusion that JA control of ABA/GA regulation in the root was time-dependent. In the root, HVED suppressed the production of 2,6 *d*HBA by 42.6% but increased SA by 77% and the BA amount by 130% with time (from the 2nd to the 5th day (Figure 4D)). The catabolism of SA includes hydroxylation reactions connected with the addition of the hydroxyl group (-OH) on three or five positions of the benzenoid ring creating sugar conjugates, 2,3-*d*HBA and 2,5-*d*HBA, respectively [68]. The LC-UV-MS *d*HBAs detection showed that 2,6-*d*HBA and 2,4-*d*HBA exhibited very similar retention times to 2,3-*d*HBA [68]. Our results reveal the positive connection between 2,6-*d*HBA and SA (*p* < 0.01, r = 0.83) (Table 3). According to this, a lower amount of 2,6-*d*HBA and higher SA and BA concentrations in 5-day-old roots suggest that SA catabolism is downregulated by HVED. In addition, an increased amount of SA and BA refers to the mobilization of the shikimate/chorismate regulatory network.

### 3.3. Shoot Hormonal and Polyphenol Response

In shoots, the early events (2 days) caused by HVED treatment were connected with reduced *d*HPA and GA20 amounts compared to the control (Figure 3A,C). The results show the synergism between ABA/GA derivates. As roots showed an antagonistic ABA/GA interaction, it can be concluded that HVED induces a tissue-specific response. In the root, ABA and GA operate together in the regulation of periclinal asymmetric cell divisions of endodermis during early root development (until 2 days) [69]. Li et al. [70] reported that in *Arabidopsis* root, a lower amount of active GA compounds was connected with higher activity of GA2ox2, a GA inactivator gene, resulting in a faster transition from cell division to cell expansion. This confirms the previous findings that the nature of ABA/GA regulation and the expression of ABA and GA coding genes are tissue-specific. On the other hand, in shoots, HVED stimulated the accumulation of the inactive prohormone JA and its active isoleucine conjugate JA_Le_Ile, revealing a positive interaction (*p*< 0.05, r = 0.82) (Table 2). HVED treatment caused mechanical changes on the seed coat which can trigger the traumatic signal followed by the accumulation of JAs. In *Arabidopsis thaliana*, mechanical injury triggered the formation of the AtPEP1, a polypeptide signal molecule, which binds to plasma membrane receptor PEPR1, resulting in the activation of JA signaling networks [71]. Decreased *d*HPA and GA20 and increased JA amounts in shoots indicate that HVED disturbs the counterbalance between the ABA and JA signaling pathways. According to Avramova [72], the type of JA and ABA crosstalk depends on the developmental stage or environment and each shift in hormonal routes causes metabolite redistribution. Metabolite analyses revealed a higher accumulation of cinnamic-acid-derived compounds (*p*-CA, CFA, BA and FA) in 2-day-old shoots after HVED exposure (Figure 4A,C). Additionally, correlation analysis confirmed a positive association between CA and FA (*p*< 0.01, r = 0.89), BA (*p* < 0.001, r = 0.95), *p*-CA (*p* < 0.05, r = 0.70), CFA (*p* < 0.05, r = 0.64) and FA (*p* < 0.05, r = 0.89), respectively (Table 2), indicating a higher induction of the phenylpropanoid pathway under HVED treatment. Similar results were obtained for *Ocimum basilicum* L. leaves in which cold plasma treatment caused a higher expression of phenylpropanoid coding genes cinnamate 4-hydroxylase (C4H), 4-coumarate coA ligase (4CL) and chavicol O-methyl transferase (CVOMT) [42]. In another study, plasma improved the phenylalanine lyase (PAL) activity of the root and shoot in *Astragalus fridae* [73]. Finally, NTP-treated rice seeds had a higher activity of enzymes related to phenylpropanoid synthesis (phenylalanine ammonia lyase, PAL) and the shikimic acid (shikimate dehydrogenase, SKDH) pathway [74].

A delayed shoot response (5th day) to HVED treatment was also characterized by higher FA production but lower BA levels (Figure 4A,C). It was reported that FA could be involved in growth promotion because of similar auxin-like properties [64]. Comparative analysis showed a negative linkage of FA with BA (*p* < 0.01, r= −0.94) and 3,4-*d*HBA (*p* < 0.05, r= −0.80), respectively (Table 2). Similarly, polyphenol content was increased in plasma-treated ginseng [75]. In addition, high levels of FA were found in the shoot after HVED treatment. Marcato et al. [76] observed that FA can absorb UV radiation through the creation of phenoxy radicals, resulting in cis-trans isomerization. Based on this, it is possible that FA might have a role in UV protection. However, further investigations are needed to prove the participation of UV in the HVED-induced accumulation of FA. In the shoot, HVED treatment stimulated JA and JA_Le_Ile production in both time points (Figure 3E). Correlation analysis for the shoot also displayed a tight association between JA metabolites on the 2nd day (*p* < 0.05, r = 0.82) and 5th day (*p* < 0.05, r = 0.94) (Table 2), confirming their role in the perception of the HVED-induced mechanical signal. The JA signaling pathway mediates plant photomorphogenesis and plays a part in the repairing of tissue damage caused by UVB [77,78]. Thus, we can assume that higher JA isoleucine accumulation could be ascribed to the developmental process or, what is more probable, the UV stress response. According to Ruan et al. [34], membrane damage activates membrane-related phospholipase which separates linoleic acid, a precursor of JA anabolism, from the phospholipid bilayer. Since the levels of JA metabolites remain elevated over time, it can be concluded that HVED in the shoot influences early and delayed JA signal transmission. Some reports even suggest the possible JA and JA-Ile resynthesis in uninjured distant tissue [79,80].

In summary, HVED treatment stimulated germination, shoot and root growth under optimal conditions. In roots, the HVED-responsible hormones were ABA and GA1, while JA and its conjugated derivate (JA_Le_Ile) were shoot-specific regulators (Figure 6). In both organs, HVED affected the phenylpropanoid pathway causing different metabolite distribution. The results show that the HVED treatment did not exert its effect through the stimulation of growth hormones (GA and IAA), but instead triggered a stress reaction, which was shown in the increase in the level of some stress hormones (e.g., JA and ABA). Altogether, the information presented here supports the idea that HVED modifies the crosstalk between the ABA, GA and JA signaling networks.

## 4. Materials and Methods

### 4.1. Plant Material and Treatment

Wheat (*Triticum aestivum* L.) cv. BC Opsesija was used in the experiments. The seeds were obtained from the BC Institute for Breeding and Production of Field Crops, Croatia. HVED treatment was applied as described previously in Marček et al. [12]. Briefly, non-sterilized seeds (30 g) were submerged in 800 mL dH_2_O and exposed to high voltage (30 kV) for 30 s and 30 Hz frequency with constant stirring. The electrical discharge was generated between the tip of the needle and the plate electrode at a distance of 1 cm. The control seeds were stirred in dH_2_O for 30 s without HVED application.

After treatment, the healthy, uniform seeds, without visible damage, were taken in plastic transparent pots (20 cm L × 15 cm W × 3 cm H) containing double-layer filter paper irrigated with 10 mL dH_2_O and covered to prevent evaporation. Seeds were left to germinate in the dark at room temperature (22 °C) and after 2 days they were exposed to fluorescent light illumination (80 mol m^−2^s^−1^) under 12 h light/dark photoperiod at 22 ± 2 °C for five days. Pots were watered (10 mL H_2_O) every second day. Each pot contained one hundred seeds and 5 pots were used for each treatment. The germination percentage was determined on 100 seeds per pot while for the shoot and root length measurements, ten plants per pot were used. Germination was evaluated visually when radicle and coleoptile were notable and expressed as germination percentages [81]. The germination was counted during the first five days whilst the shoot and root growth were recorded from the 2nd to the 5th day using a scaler ruler.

### 4.2. Hormone and Metabolite Measurements

For the hormone and metabolite determination, 200 mg shoot and root tissues were collected (200 mg per sample) from each pot from the 2nd to 5th day of germination. The 5 pots of each treatment provided the biological repetitions (n = 5). The samples were stored at −80 °C until analysis.

Samples were (200 mg) homogenized using a Qiagen TissueLyser II (Hilden, Germany) and extracted in 1 mL of 75% (*v*/*v*) methanol (UPLC gradient grade; VWR, Radnor, PA, USA) using a 1600 MiniG^®^-Automated Tissue Homogenizer and Cell Lyser (SPEX; Rickmansworth, UK) for 3 min at 1250 RPM. Twenty ng/mL [2H2]GA4 and [2H4]GA1 as internal standards were added to each sample. Then, samples were centrifuged at 16,500 g for 15 min at +4 °C and the extraction process was repeated. The two supernatants were pooled and partitioned against 1 mL diethyl-ether: petrolether = 4:1 (*v*/*v*%) (HPLC grade; VWR). After phase separation, the lower (methanol–water) phase was subjected to flow through type SPE clean-up using an ENVI-18 SPE tube (1 mL; Supelclean, Merck-Sigma Group; Schnelldorf, Germany). After clean-up, the final sample extract was evaporated under vacuum at 35 °C and reconstituted in 400 µL of 30% (*v*/*v*) methanol containing 0.1% (*v*/*v*) formic acid (MS grade; VWR). Then, the samples were filtered through a 0.22 µm disposable PTFE syringe filter and immediately submitted for UPLC-US-MS/MS analysis. The UPLC separation was performed using a Waters Acquity I class UPLC system (Milford, MA, USA) and a Waters Acquity HSS T3 column (1.8 μm, 100 mm × 2.1 mm) at 40 °C. The mobile phase was composed of a mixture of 0.1 *v*/*v*% formic acid (FA) (A) and acetonitrile containing 0.1 *v*/*v* % FA (B), according to Appendix A. Samples of 1.5 μL were injected. Tandem mass spectrometric detection was performed on a Waters Xevo TQ-XS equipped with a UniSpray™ source (US) operated in timed multiple reaction monitoring (MRM) mode. The setting parameters are described in Appendix A. Where possible, at least three MRM transitions were used for data acquisition and the transition having the highest S/N ratio was used for quantitation (Appendix A). Calibration was carried out via external procedural calibration points and data processing was conducted using Waters MassLynx 4.2 and TargetLynx software.

### 4.3. Data Analysis

Statistical analyses were performed using Statistica 14.0.0.15 (TIBCO Software Inc., Palo Alto, CA, USA). The results in this study were expressed as the mean value ± standard deviation of 3–5 repetitions. To check the variability of the results and the interactions of treatment (T), day (D) and combined T × D, the factorial analysis of variance (ANOVA) was performed. Differences among the variables (T and D) were compared using Fisher’s least significant difference (LSD) test. To reveal the associations between variables, Spearman’s correlation coefficients were calculated in two time points (2nd and 5th day) for the root and shoot, separately. Principal component analysis (PCA) was applied to evaluate and discriminate the hormonal and metabolic responses of the shoot and root between different treatments during the time (2nd and 5th day). The data set for PCA consisted of 21 variables for the root and 22 variables for the shoot, respectively. Factor loadings were performed to assess the proportion of total variance with different principal components (PCs). The loadings showed the correlations between different PC and variables, whereas higher values (>0.71) were referred to as strong [82].

## Figures and Tables

**Figure 1 plants-12-01235-f001:**
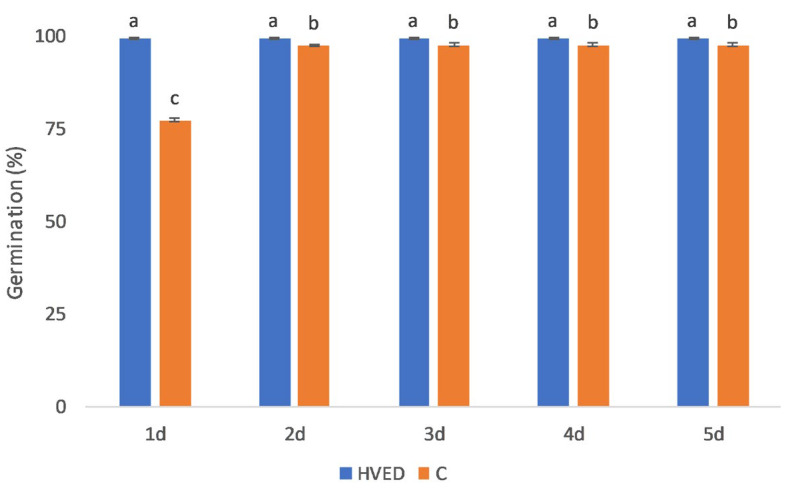
Germination percentage under control and high voltage electrical discharge (HVED) treatment across five days. Values are means of five repetitions ± S.D. The different letters denote significances among the means between days and treatments at *p* < 0.05 using the LSD post hoc test.

**Figure 2 plants-12-01235-f002:**
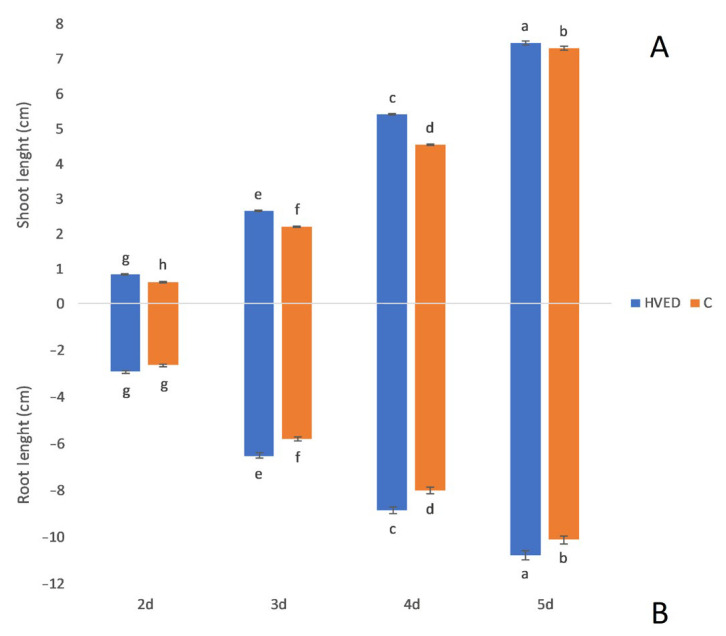
Shoot (**A**) and root (**B**) length under control C and high voltage electrical discharge (HVED) treatment for the 2nd and 5th day. Values are means of five repetitions ± S.D. The different letters denote significances among means between days and treatments at *p* < 0.05 using the LSD post hoc test. C- denotes control group.

**Figure 3 plants-12-01235-f003:**
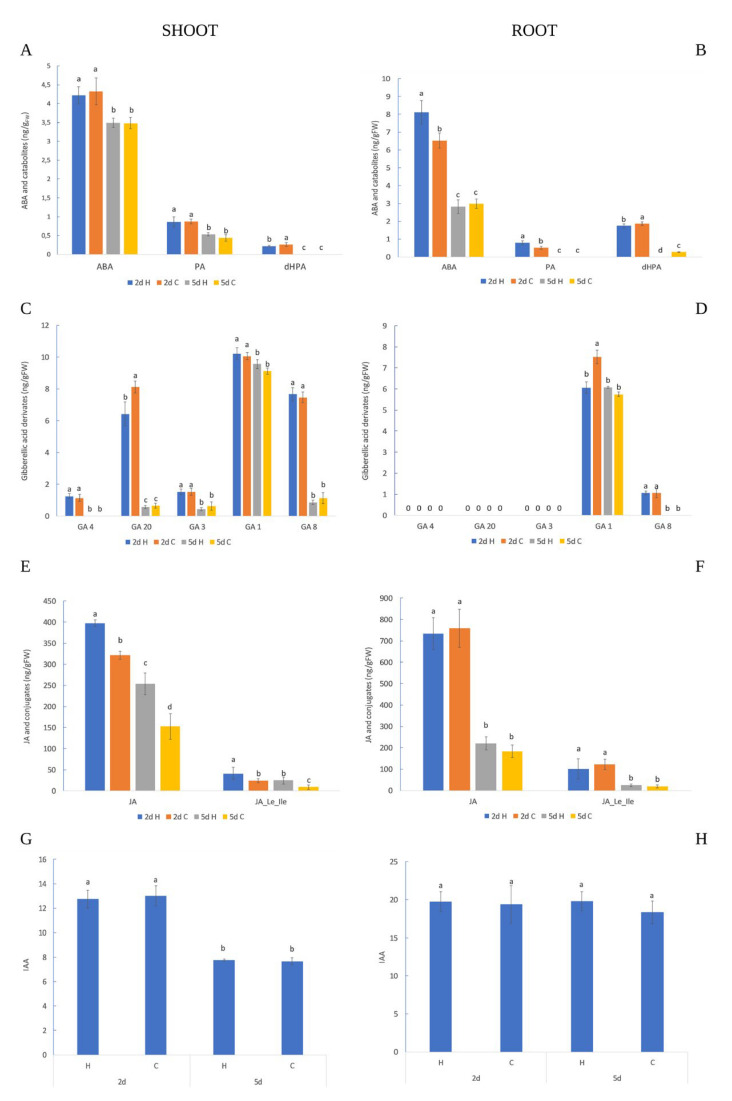
Amount of hormones in shoot (**A**,**C**,**E**,**G**) and root (**B**,**D**,**F**,**H**) under control (**C**) and high electrical voltage discharge (HVED) treatment (H) on the 2nd and 5th day, respectively. Abscisic acid (ABA); phaseic acid (PA); dihydrophaseic acid (*d*HPA); gibberellic acid (GA) derivates; jasmonic acid (JA); jasmonoyl-leucine-isoleucine (JA_Le_Ile) and auxin (IAA). Values are means of 3–5 repetitions ± S.D. The different letters denote significances among means between days and treatments at *p* < 0.05 using the LSD post hoc test.

**Figure 4 plants-12-01235-f004:**
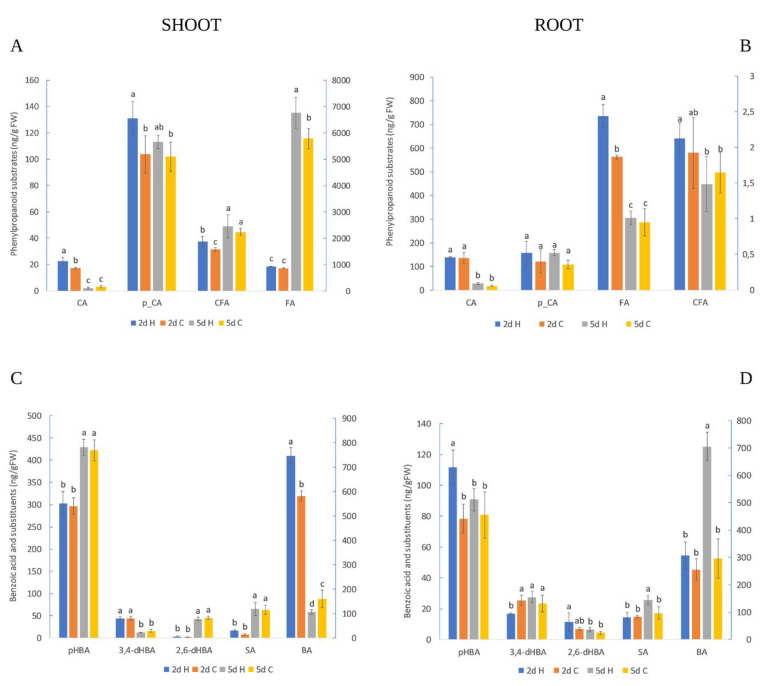
Amount of polyphenols in shoot (**A**,**C**) and root (**B**,**D**) under control (**C**) and high electrical voltage discharge (HVED) treatment (H) on 2nd and 5th day, respectively. In 4A, the left scale shows the amount of CA, *p*-CA and CFA, and the right scale shows FA. In 4B, the left scale shows CA, *p*-CA and FA, and the right scale presents CFA. Cinnamic acid (CA); *p*-coumaric acid (*p*-CA); ferulic acid (FA); caffeic acid (CFA); *p*-hydroxybenzoic acid (*p*-HBA); 3,4-dihydroxybenzoic acid (3,4-*d*HBA); 2,6-dyhydroxybenzoic acid (2,6-*d*HBA); salicylic acid (SA) and benzoic acid (BA). Values are means of 3–5 repetitions ± S.D. The different letters denote significances among means between days and treatments at *p* < 0.05 using the LSD post hoc test.

**Figure 5 plants-12-01235-f005:**
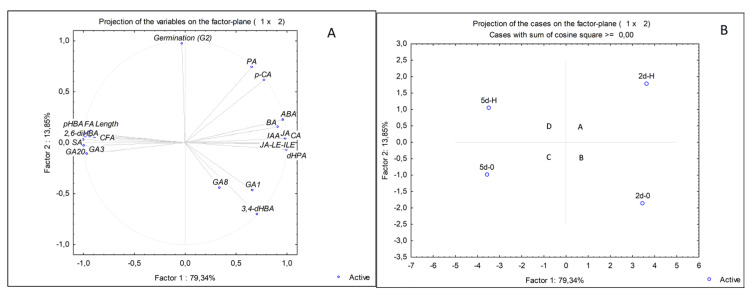
Principal component analysis of data sets of hormone and polyphenol accumulation in the root under control (0) and high voltage electrical discharge (HVED) treatment on the 2nd and 5th day. Factor loading projection (**A**) and scores (**B**) of two principal components. The capital letters denote groups. Abbreviations: abscisic acid (ABA); auxin (IAA); benzoic acid (BA); caffeic acid (CFA); cinnamic acid (CA); 3,4- dihydroxybenzoic acid (3,4-*d*HBA); 2,6-dihydroxybenzoic acid (2,6-*d*HBA); dihydrophaseic acid (*d*HPA); ferulic acid (FA); gibberellic acid derivates (GAs); high voltage electrical discharge (HVED); 4-hydroxybenzoic acid (*p*-HBA); jasmonic acid (JA); jasmonoyl-leucine-isoleucine (JA_Le_Ile); *p*-coumaric acid (*p*-CA); phaseic acid (PA); salicylic acid (SA).

**Figure 6 plants-12-01235-f006:**
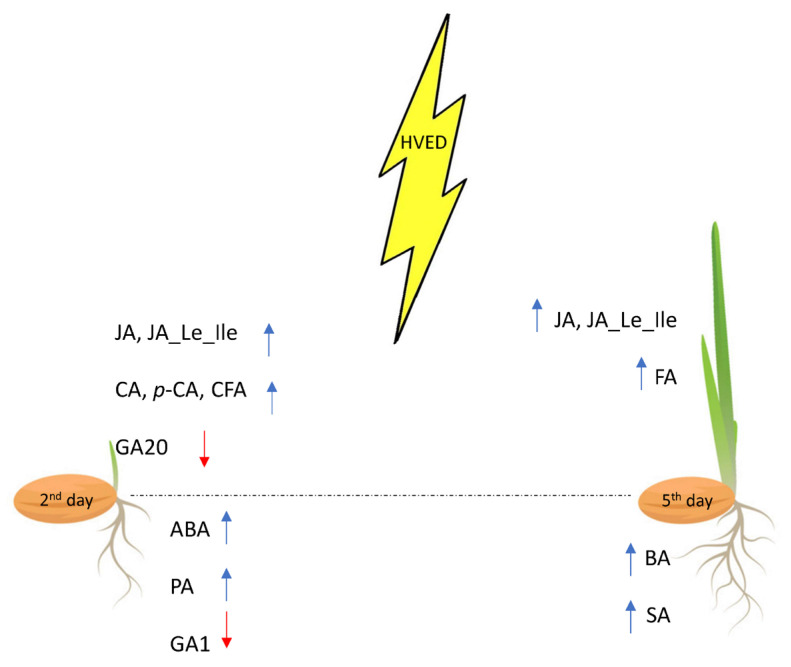
Time-dependent HVED impact on hormonal and phenylpropanoid distribution in the root and shoot. Blue arrows denote an increase, and red arrows denote a decrease.

**Table 1 plants-12-01235-t001:** Spearman’s correlation coefficients among germination (G), shoot (S) and root (R) length under control and high voltage electrical discharge (HVED) treatment (*, **, ***: significant at 0.05, 0.01 and 0.001 level). Numbers beside capital letters represent the treatment day.

Variable	Treatment	S2	S3	S4	S5	R2	R3	R4	R5	G1	G2	G3
Treatment												
S2	0.91 ***											
S3	0.90 ***	0.82 ***										
S4	0.89 ***	0.86 ***	0.82 ***									
S5	0.89 ***	0.75 ***	0.80 ***	0.68 *								
R2	0.90 ***	0.83 ***	0.76 ***	0.85 ***	0.74 ***							
R3	0.89 ***	0.84 ***	0.70 ***	0.77 ***	0.69 ***	0.82 ***						
R4	0.88 ***	0.81 ***	0.75 ***	0.78 ***	0.66 ***	0.83 ***	0.86 ***					
R5	0.77 ***	0.78 ***	0.59 *	0.57 *	0.68 ***	0.75 ***	0.78 ***	0.74 ***				
G1	0.88 ***	0.78 ***	0.84 ***	0.81 ***	0.77 ***	0.80 ***	0.74 ***	0.75 ***	0.65 *			
G2	0.81 ***	0.70 ***	0.78 ***	0.78 ***	0.62 *	0.72 ***	0.74 ***	0.77 ***	0.58 *	0.85 ***		
G3	0.80 ***	0.70 ***	0.79 ***	0.77 ***	0.64 *	0.73 ***	0.71 ***	0.75 ***	0.53 **	0.86 ***	0.97 ***	

**Table 2 plants-12-01235-t002:** Spearman’s correlation coefficients among germination, shoot length, hormones and their derivates under control and high voltage electrical discharge (HVED) treatment after the 2nd and 5th day of growth (*, **, ***: significant at 0.05, 0.01 and 0.001 level).

5 DAY
	Variable	Treatment	CA	IAA	BA	*p*HBA	SA	3,4-*d*HBA	2,6-*d*HBA	*p*-CA	CFA	FA	JA	ABA	PA	*d*HPA	JA-Le-Ile	GA 4	GA 20	GA 3	GA 1	GA 8	G%	ShL
2 DAY	Treatment		−0.58 ns	0.06 ns	−0.87 *	0.28	0.17 ns	−0.73 *	−0.39 ns	0.51 ns	0.28 ns	0.68 ns	0.88 *	−0.06 ns	0.51 ns	*n.d.*	0.85 **	*n.d.*	−0.28 ns	−0.51 ns	0.88 *	−0.28 ns	0.80 ***	0.88 ***
CA	0.87 **		−0.14 ns	0.36 ns	0.57 ns	0.21 ns	0.43 ns	−0.21 ns	0.46 ns	0.11 ns	−0.20 ns	−0.60 ns	−0.05 ns	−0.25 ns	−0.57 ns	0.04 ns	0.32 ns	−0.37 ns	0.39 ns	−0.44 ns	−0.47 ns
IAA	−0.25 ns	−0.27 ns		−0.61 ns	−0.33 ns	0.02 ns	−0.19 ns	0.21 ns	−0.12 ns	−0.26 ns	0.71 ns	0.89 *	0.20 ns	0.40 ns	−0.29 ns	−0.26 ns	−0.21 ns	−0.03 ns	0.05 ns	−0.14 ns	0.01 ns
BA	0.87 **	0.95 ***	0.00 ns		−0.32 ns	−0.54 ns	0.93 **	0.21 ns	−0.46 ns	−0.46 ns	−0.94 **	−0.77 ns	0.23 ns	−0.54 ns	−0.61 ns	0.57 ns	0.57 ns	−0.83 *	0.43 ns	−0.65 ns	−0.67 ns
*p*HBA	0.14 ns	0.10 ns	−0.14 ns	0.04 ns		0.45 ns	−0.45 ns	−0.45 ns	0.74 *	0.43 ns	0.60 ns	−0.03 ns	−0.61 ns	0.00 ns	0.48 ns	−0.45 ns	−0.14 ns	0.37 ns	−0.36 ns	0.17 ns	0.06 ns
SA	0.87 **	0.69 ns	−0.22 ns	0.60 ns	0.23 ns		−0.64 ns	0.26 ns	0.24 ns	0.90 **	0.71 ns	−0.03 ns	−0.60 ns	−0.52 ns	−0.02 ns	−0.69 ns	−0.48 ns	0.20 ns	−0.62 ns	0.10 ns	0.08 ns
3,4-*d*HBA	−0.03 ns	0.22 ns	−0.70 *	0.12 ns	0.18 ns	−0.19 ns		−0.07 ns	−0.38 ns	−0.55 ns	−0.8 *	−0.71 ns	0.42 ns	−0.02 ns	−0.55 ns	0.71 *	0.55 ns	−0.60 ns	0.71 *	−0.45 ns	−0.50 ns
2,6-*d*HBA	0.76 *	0.67 ns	0.02 ns	0.66 ns	0.05 ns	0.71 ns	−0.31 ns		−0.76 *	0.14 ns	−0.09 ns	−0.49 ns	−0.28 ns	−0.55 ns	−0.52 ns	−0.14 ns	0.17 ns	−0.49 ns	−0.45 ns	−0.41 ns	−0.26 ns
*p*-CA	0.69 *	0.70 *	0.29 ns	0.86 *	0.50 ns	0.71 ns	−0.10 ns	0.52 ns		0.31 ns	0.43 ns	0.26 ns	−0.01 ns	0.36 ns	0.45 ns	−0.10 ns	−0.12 ns	0.49 ns	0.17 ns	0.49 ns	0.44 ns
CFA	0.66 *	0.64 *	−0.21 ns	0.57 ns	0.78 **	0.71 *	0.13 ns	0.60 ns	0.70 *		0.54 ns	−0.09 ns	−0.54 ns	−0.48 ns	0.10 ns	−0.40 ns	−0.33 ns	0.49 ns	−0.48 ns	0.38 ns	0.34 ns
FA	0.87 *	0.89 *	−0.11 ns	0.94 **	0.14 ns	0.60 ns	0.00 ns	0.49 ns	0.94 *	0.57 ns		0.60 ns	−0.54 ns	0.54 ns	0.54 ns	−0.89 *	−0.37 ns	0.90 *	−0.66 ns	0.62 ns	0.60 ns
JA	0.87 *	0.99 ***	−0.11 ns	0.93 **	0.20 ns	0.66 ns	0.36 ns	0.70 ns	0.77 ns	0.82 *	0.90 *		0.09 ns	0.94 **	0.94 *	−0.60 ns	−0.77 ns	0.70 ns	−0.26 ns	0.62 ns	0.54 ns
ABA	0.00 ns	0.30 ns	−0.13 ns	0.14 ns	0.33 ns	0.21 ns	0.10 ns	0.36 ns	0.05 ns	0.42 ns	0.20 ns	0.43 ns		0.29 ns	−0.22 ns	0.59 ns	0.14 ns	−0.26 ns	0.78 *	0.00 ns	0.09 ns
PA	−0.24 ns	−0.08 ns	0.25 ns	−0.02 ns	−0.04 ns	0.05 ns	0.14 ns	0.05 ns	−0.05 ns	−0.02 ns	−0.36 ns	0.14 ns	0.20 ns		0.43 ns	0.24 ns	0.05 ns	0.71 ns	0.38 ns	0.49 ns	0.50 ns
*d*HPA	−0.35 ns	−0.43 ns	0.37 ns	−0.39 ns	−0.17 ns	−0.32 ns	−0.65 ns	−0.12 ns	−0.10 ns	−0.30 ns	−0.14 ns	−0.54 ns	−0.33 ns	−0.62 ns		*n.d.*	*n.d.*				
JA-Le_Ile	0.59 ns	0.73 *	−0.35 ns	0.71 *	−0.18 ns	0.74 *	0.03 ns	0.86 **	0.37 ns	0.28 ns	0.54 ns	0.82 *	0.33 ns	−0.09 ns	−0.02 ns		−0.29 ns	−0.43 ns	0.77 ns	−0.36 ns	0.68 ns	0.56 ns
GA 4	0.38 ns	0.13 ns	−0.52 ns	−0.07 ns	−0.35 ns	0.17 ns	−0.01 ns	0.19 ns	−0.35 ns	−0.09 ns	−0.07 ns	−0.18 ns	−0.33 ns	−0.55 ns	0.17 ns	0.19 ns		*n.d.*
GA 20	−0.48 ns	−0.47 ns	0.46 ns	−0.71 ns	−0.35 ns	−0.36 ns	−0.38 ns	−0.02 ns	−0.22 ns	−0.48 ns	−0.43 ns	−0.71 ns	−0.54 ns	0.36 ns	0.34 ns	−0.06 ns	−0.21 ns		0.76 *	0.03 ns	0.81 *	0.10 ns	0.14 ns
GA 3	0.10 ns	0.02 ns	−0.29 ns	−0.05 ns	0.35 ns	−0.02 ns	−0.04 ns	0.29 ns	0.13 ns	0.16 ns	0.32 ns	−0.07 ns	0.30 ns	−0.42 ns	0.10 ns	0.18 ns	−0.02 ns	−0.13 ns		−0.37 ns	0.48 ns	−0.22 ns	−0.12 ns
GA 1	0.30 ns	0.37 ns	−0.40 ns	0.67 ns	0.22 ns	−0.45 ns	0.89 **	−0.30 ns	0.37 ns	0.11 ns	0.67 ns	0.63 ns	−0.41 ns	−0.19 ns	−0.48 ns	−0.37 ns	0.07 ns	−0.39 ns	−0.11 ns		−0.09 ns	0.93 **	0.84 *
GA 8	0.24 ns	0.33 ns	−0.13 ns	0.17 ns	0.81 **	0.02 ns	0.35 ns	0.43 ns	0.60 ns	0.76 *	0.32 ns	0.43 ns	0.38 ns	0.10 ns	−0.45 ns	−0.08 ns	−0.38 ns	−0.34 ns	0.38 ns	0.48 ns		−0.01 ns	0.01 ns
G%	0.81 ***	0.71 *	−0.17 ns	0.75 *	0.20 ns	0.82 *	−0.08 ns	0.53 ns	0.70 *	0.61 ns	0.73 ns	0.65 ns	−0.14 ns	−0.23 ns	−0.34 ns	0.41 ns	0.32 ns	−0.64 ns	−0.06 ns	0.32 ns	0.09 ns		0.77 **
ShL	0.92 ***	0.85 **	−0.15 ns	0.91 **	0.32 ns	0.70 ns	0.01 ns	0.60 ns	0.83 **	0.69 *	0.95 ***	0.92 **	0.19 ns	−0.34 ns	−0.27 ns	0.47 ns	0.13 ns	−0.65 ns	0.24 ns	0.39 ns	0.40 ns	0.83 ***	
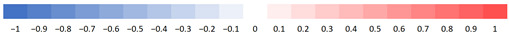

*Abbreviations*: CA (cinnamic a.); *p*-CA (*p*-coumaric a.); CFA (caffeic a.); FA (ferulic a.); PA (phaseic a.); ABA (abscisic acid); PA (phaseic acid); *d*HPA (dihydrophaseic acid); JA-Le_Ile (jasmonoyl-leucine-isoleucine); IAA (auxin); *p*HBA (*p*-hydroxybenzoic acid); 3,4-*d*HBA (3,4-dihydroxybenzoic acid); 2,6-*d*HBA (2,6-dihydroxybenzoic acid); SA (salicylic acid); BA (benzoic acid); GA (gibberelins 1,3,4,8,20); JA (jasmonic acid); G% (germination percentage); ShL (shoot length); n.d. (not detected).

**Table 3 plants-12-01235-t003:** Spearman’s correlation coefficients among germination, root length, hormones and their derivates under control and high voltage electrical discharge (HVED) treatment after the 2nd and 5th day of growth (*, **, ***: significant at 0.05, 0.01 and 0.001 level).

5 DAY
	Variable	Treatment	CA	IAA	BA	*p*HBA	SA	3,4-*d*HBA	2,6-*d*HBA	*p*-CA	CFA	FA	JA	ABA	PA	*d*HPA	JA-Le_Ile	GA 1	GA 8	G%	RhL
2 DAY	Treatment		0.87 **	0.52 ns	0.87 **	0.52 ns	0.69 *	0.43 ns	0.78 *	0.61	−0.17 ns	0.22 ns	0.65 ns	−0.26 ns	*n.d.*	−0.90 ***	0.43 ns	0.87 *	*n.d.*	0.81 **	0.18 ns
CA	0.10 ns		0.41 ns	0.78 *	0.53 ns	0.68 *	0.30 ns	0.64 ns	0.72 *	−0.04 ns	0.36 ns	0.63 ns	−0.46 ns	−0.89 **	0.09 ns	0.56 ns	0.62 ns	−0.08 ns
IAA	0.09 ns	−0.32 ns		0.60 ns	0.07 ns	0.47 ns	0.90 **	0.48 ns	0.22 ns	−0.12 ns	0.29 ns	0.33 ns	−0.30 ns	−0.52 ns	0.35 ns	0.25 ns	0.47 ns	0.62 ns
BA	0.58 ns	0.40 ns	0.50 ns		0.62 ns	0.74 *	0.36 ns	0.76 *	0.60 ns	−0.24 ns	0.25 ns	0.11 ns	0.12 ns	−0.81 *	0.31 ns	0.75 ns	0.69 ns	0.00 ns
*p*HBA	0.87 **	0.26 ns	−0.22 ns	0.36 ns		0.30 ns	−0.14 ns	0.27 ns	0.83 **	0.18 ns	0.57 ns	0.24 ns	−0.28 ns	−0.30 ns	−0.08 ns	0.71 ns	0.50 ns	−0.52 ns
SA	0.17 ns	−0.67 ns	0.11 ns	0.58 ns	0.20 ns		0.55 ns	0.83 **	0.50 ns	0.00 ns	0.40 ns	0.26 ns	0.03 ns	−0.64 ns	0.23 ns	0.21 ns	0.45 ns	0.28 ns
3,4-dHBA	−0.88 **	−0.41 ns	−0.01 ns	−0.31 ns	−0.78 *	0.04 ns		0.54 ns	0.08 ns	−0.13 ns	0.14 ns	0.40 ns	−0.28 ns	−0.40 ns	0.44 ns	0.14 ns	0.41 ns	0.81 **
2,6-dHBA	0.26 ns	0.03 ns	0.42 ns	0.64 ns	0.08 ns	0.30 ns	−0.20 ns		0.47 ns	−0.28 ns	0.14 ns	0.38 ns	0.02 ns	−0.71 *	0.58 ns	0.64 ns	0.57 ns	0.44 ns
*p*-CA	0.49 ns	0.40 ns	0.77 ns	0.54 ns	0.31 ns	−0.40 ns	−0.70 ns	0.31 ns		0.42 ns	0.38 ns	0.55 ns	−0.52 ns	−0.41 ns	0.15 ns	0.54 ns	0.47 ns	−0.22 ns
CFA	0.35 ns	0.88 *	−0.43 ns	−0.05 ns	0.54 ns	−0.46 ns	−0.59 ns	−0.11 ns	0.37 ns		−0.33 ns	0.14 ns	−0.30 ns	0.37 ns	0.02 ns	−0.29 ns	−0.10 ns	−0.21 ns
FA	0.88 *	−0.30 ns	0.83 *	0.49 ns	0.66 ns	0.20 ns	−0.60 ns	0.60 ns	0.8 *	0.06 ns		−0.11 ns	−0.05 ns	−0.43 ns	−0.62 ns	−0.25 ns	0.01 ns	−0.23 ns
JA	0.00 ns	0.20 ns	−0.86 *	−0.43 ns	0.21 ns	0.43 ns	−0.31 ns	−0.36 ns	−0.89 *	0.36 ns	−0.90 *		−0.83 *	−0.46 ns	0.48 ns	0.37 ns	0.70 ns	0.06 ns
ABA	0.88 *	0.50 ns	0.09 ns	0.70 ns	0.77 ns	0.10 ns	−0.80 ns	0.71 ns	0.20 ns	0.37 ns	0.8 *	0.14 ns		0.18 ns	−0.05 ns	−0.07 ns	−0.28 ns	−0.05 ns
PA	0.87 **	0.26 ns	0.02 ns	0.32 ns	0.72 *	0.30 ns	−0.92 **	0.28 ns	0.31 ns	0.47 ns	0.83 *	0.36 ns	0.83 *		*n.d.*	*n.d.*
*d*HPA	−0.49 ns	0.40 ns	−0.89 *	−0.60 ns	−0.20 ns	0.20 ns	0.52 ns	−0.37 ns	−0.90 *	0.20 ns	−0.98 **	0.90 *	−0.20 ns	−0.37 ns		−0.22 ns	−0.79 *	−0.68 *	−0.08 ns
JA-LE_Ile	−0.22 ns	−0.26 ns	0.12 ns	0.22 ns	−0.25 ns	0.61 ns	0.57 ns	0.51 ns	−0.55 ns	−0.47 ns	0.35	0.05 ns	0.03 ns	−0.04 ns	0.38 ns		0.68 ns	0.53 ns	0.56 ns
GA 1	−0.88 *	0.50 ns	−0.60 ns	−0.20 ns	−0.66 ns	0.00 ns	0.60 ns	−0.60 ns	−0.8 *	−0.06 ns	−0.83 *	0.90 *	−0.60 ns	−0.66 ns	0.99 ***	0.03 ns		0.89**	−0.09 ns
GA 8	0.26 ns	0.32 ns	−0.56 ns	−0.07 ns	0.38 ns	0.31 ns	−0.36 ns	0.27 ns	−0.60 ns	0.45 ns	−0.14 ns	0.86 *	0.31 ns	0.57 ns	0.70 ns	0.36 ns	0.20 ns		*n.d.*
G%	0.8 **	0.31 ns	−0.01 ns	0.64 ns	0.78 *	−0.01 ns	−0.74 *	0.42 ns	0.27 ns	0.33 ns	0.74 ns	−0.04 ns	0.99 ***	0.71 *	−0.21 ns	−0.11 ns	−0.74 ns	0.25 ns		0.30 ns
RhL	0.90 ***	0.19 ns	−0.20 ns	0.16 ns	0.82 **	−0.16 ns	−0.85 **	−0.04 ns	0.38 ns	0.57 ns	0.79 *	0.24 ns	0.79 *	0.85 **	−0.26 ns	−0.45 ns	−0.88 *	0.33 ns	0.69 *	
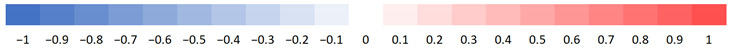

*Abbreviations*: CA (cinnamic a.); *p*-CA (*p*-coumaric a.); CFA (caffeic a.); FA (ferulic a.); PA (phaseic a.); ABA (abscisic acid); PA (phaseic acid); *d*HPA (dihydrophaseic acid); JA-Le_Ile (jasmonoyl-leucine-isoleucine); IAA (auxin); *p*HBA (*p*-hydroxybenzoic acid); 3,4-dHBA (3,4-dihydroxybenzoic acid); 2,6-*d*HBA (2,6-dihydroxybenzoic acid); SA (salicylic acid); BA (benzoic acid); GA (gibberelins 1,3,4,8,20); JA (jasmonic acid); G% (germination percentage); RhL (root length); not detected (n.d.).

## Data Availability

Not applicable.

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
