# Peer review of "Effects of High Voltage Electrical Discharge (HVED) on Endogenous Hormone and Polyphenol Profile in Wheat"

_plants, 2023, doi:10.3390/plants12061235_

Round 1
Reviewer 1 Report
The main goal of the study was to determine High Voltage Electrical Discharge (HVED) on the hormonal and metabolic changes during the germination of wheat seedlings. The Authors detected the hormonal changes including abscisic acid (ABA), gibberellic acids (GAs), indol acetic acid (IAA), jasmonic acid (JA) and polyphenol responses were detected on early (2nd day) and late (5th day) germination phase of wheat and their redistribution in shoot and root. The Authors showed interesting results. HVED treatment significantly stimulated germination and growth both in the shoot and root. HVED- provoked metabolic changes indicated a stress-related response that can contribute germination in wheat. Work is written correctly, research methods selected correctly as well, results presented in a clear manner, enough literature included, conclusions are answered on the main goals.
Please use the correct spelling of units and scripts as well.
After all corrections marked in text the paper can be publish in the MDPI Plants Journal.

Author Response
Please use the correct spelling of units and scripts as well.
ANSWER: Thank you very much for this suggestion. They are corrected.
After all corrections marked in text the paper can be publish in the MDPI Plants Journal.
ANSWER: We sincerely appreciate the time you've spent on manuscript revision.

Reviewer 2 Report
The manuscript "Effects of high voltage electrical discharge (HVED) on endogenous hormone and polyphenol profile in wheat" by Tihana Marček, Kamirán Áron Hamow, Tibor Janda, Eva Darko presents a study on metabolic changes indicated a stress-related response that can contribute germination in wheat with a special perspective on the synthesis of bioactive gibberellins. A large volume of work is unfortunately poorly structured and raises many questions. This makes it difficult to evaluate this work on its merits.
My remarks:
The article is very large in terms of the amount of data presented, at the same time data on germination, 2 growth indicators and 20 physiological indicators (11 hormones and their derivatives and 9 polyphenols) are presented. This makes the article very difficult to read. Perhaps the authors should split it into two articles - 1) the effect of HVED treatment of wheat seeds on hormone levels in wheat roots and seedlings; 2) the effect of HVED treatment of wheat seeds on the content of phenolcarboxylic acids in wheat roots and seedlings. In my opinion, these data are not related in meaning.
Methodological remarks - why were the 2nd and 5th days chosen for the analyzes of the content of hormones and polyphenols? How many replicates were sampled for these analyzes - p. 609 "on the 2nd to 5th day of germination in five repetitions", p. 220 and 254 "Values are means of 3-5 repetitions"? Check out the Materials and Methods section.
Are the calculated correlation coefficients reliable at P<0.05, P<0.01, P<0.001 probabilities with such a number of repetitions? Check out the Materials and Methods section.
The discussion of the results. When discussing the results, there is not enough tabular data on the content of hormones and polyphenols in wheat roots and seedlings (they need to be added to the additional materials section), since these data presented in the form of graphs are very difficult to understand. I would like the authors to use the same color scheme on all graphs to depict the variants of the experience - on graphs 1 and 2, blue means control, orange means HVED. And on graphs 3 and 4 - the opposite is true.
The authors have a very complex style of presenting their findings: p. 198 "The biosynthesis of JA and JA_Le_Ile decreased over time (Figures 3e and 3f)." and adjacent pp. 201-204 "The JA concentration in 2-day-old root was 314% (control group) and 234% higher (HVED group) than in 5-day-old root while the amount of JA-Ile derivates increased by 535% and 304%, respectively (Figure 3f)." It may be worth writing that in 5-day-old seedlings the content of these indicators is only lower than in two-day-old seedlings. 235-237 "At the beginning (2nd day), 3,4-dHBA amounts raised while later increase was noticed for SA, pHBA and 236 2,6-dHBA, respectively." means that the content of 3,4-dHBA decreased from the second to the fifth day, while the content of other compounds increased!! How can this be explained or is it a bug? This needs to be corrected.
Tables 2 and 3 present a very large percentage of statistically insignificant results. Why is this done? It is not indicated what "-" means.
In chapter 2.4 fig. 5 is completely unreadable. Also, the question arises why this chapter is given and Principal components analysis is done if it is not discussed further. This must either be supplemented or removed from the manuscript.
The chapter "Discussion" also contains a large amount of material, but in my opinion it is very difficult to understand. Some conclusions made by the authors are highly controversial – pp. 476-482 “Our results showed a higher FA (Ferulic acid) amount in 2-day-old-HVED roots which might be connected with the modifications of the cell walls. Calcium ions are especially important in maintaining cell wall rigidity…. In this context, it seems that HVED treatment has the potential in mediating the Ca2+ dependent processes." Moreover, by the fifth day, the FA content in the roots decreases sharply!!
The work requires careful processing and then again for review.
Author Response
"Please see the attachment."

Reviewer 3 Report
Dear Authors, the article is interesting and well written, being an interest topic for the agriculture.
This research studied the mode of action of d high voltage electrical discharge (HVED) treatment on germination and seedling development (on the 2nd and 5th day) and to follow the alterations in hormone profile (ABA, GAs, IAA, JA) and their derivates, phenylpropanoids, and benzoic acids in wheat.
Considering the field of research of this journal, the work is very valuable.
The experimental part is accurately done and very complex, and results are well-presented bringing to solid discussions. In the attached file authors can find some comments I did on the manuscript. I appreciate that references are very recent.
Below are my remarks and comments.
The manuscript do not respect the journal template regarding the ref in the text. Have to be [xx], not [xx]. Also the authors should justify the text. And I don’t understand why the template is from 2021. Please revise it.
For all Figures: It can be delete „Data were analysed by using STATISTICA 14.0.0.15 software package.” and mentioned only at Statistic Section.
Overall I think this is a very good article, with important and a high volume of scientific information and I want to congratulate the authors.
Author Response
Dear Authors, the article is interesting and well written, being an interest topic for the agriculture.
This research studied the mode of action of d high voltage electrical discharge (HVED) treatment on germination and seedling development (on the 2nd and 5th day) and to follow the alterations in hormone profile (ABA, GAs, IAA, JA) and their derivates, phenylpropanoids, and benzoic acids in wheat.
Considering the field of research of this journal, the work is very valuable.
The experimental part is accurately done and very complex, and the results are well-presented bringing to solid discussions. In the attached file authors can find some comments I did on the manuscript. I appreciate that the references are very recent.
ANSWER: The authors thank the reviewer for the positive response and valuable comments. Despite all our efforts, we could not find the attached file, but we could correct the remarks below. The oldest reference dates from 2001 (Yukiko et al.). We leave this reference and also add Marchiosi et al. (2020). (line:515)
Below are my remarks and comments.
The manuscript do not respect the journal template regarding the ref in the text. Have to be [xx], not [xx]. Also the authors should justify the text. And I don’t understand why the template is from 2021. Please revise it.
ANSWER: Thank you for your comment. We did the corrections throughout the manuscript. The text is justified and we apologize for using the wrong template. In the revised version, the manuscript is formatted according to the template 2023. Thank you very much for calling our attention to the mistakes.
For all Figures: It can be delete „Data were analysed by using STATISTICA 14.0.0.15 software package.” and mentioned only at Statistic Section.
ANSWER: Thank you for your comment. We removed text under Figures and from Supplementary Material also.
Overall I think this is a very good article, with important and a high volume of scientific information and I want to congratulate the authors.
ANSWER: Thank you very much for your comments. We sincerely appreciate the time you've spent on manuscript revision.
Reviewer 4 Report
Authors investigated the affects of applying HVED on wheat seeds to promote germination, hormonal and metabolic activity. The experiment was well described in all aspects and the Authors clearly presented the methodology, the results and they also provide exhaustive explanation of the phenomena correlated with the treatment applied. Conclusions are consistent with results.
Some specific comments are provided below to further improve the article before publication:
[146-147] check comma in numbers
Figure 1: graph font
Figure 2: I suggest to use capital letters for Treatments to avoid misunderstanding with significance
Table 1 please check commas
Having described the software used for statistical analysis in the dedicated section (4.3 data analysis), it is preferable to remove the repetition after graphs and tables
Reference numbers within the test are reported as superscript.
Editing and test formatting needed
Author Response
Reviewer 4.
Authors investigated the effects of applying HVED on wheat seeds to promote germination, hormonal and metabolic activity. The experiment was well described in all aspects and the Authors clearly presented the methodology, and the results and they also provide an exhaustive explanation of the phenomena correlated with the treatment applied. Conclusions are consistent with results.
Some specific comments are provided below to further improve the article before publication:
[146-147] check comma in numbers
ANSWER: Thank you very much for your comment. We changed the commas into dots throughout the manuscript including Table 1 and Supplementary Material.
Figure 1: graph font
ANSWER: Yes, we changed it.
Figure 2: I suggest to use capital letters for Treatments to avoid misunderstanding with significance
ANSWER: Thank you very much for the suggestion. The letters are changed to capital in Figs 2,3,4, and 5 and through text.
Table 1 please check commas
ANSWER: We changed them.
Having described the software used for statistical analysis in the dedicated section (4.3 data analysis), it is preferable to remove the repetition after graphs and tables
ANSWER: Description is removed from each figure and table.
Reference numbers within the test are reported as superscript.
ANSWER: Sorry for it, they are corrected.
Editing and test formatting needed
ANSWER: Thank you very much for calling our attention to the mistakes. We corrected the formatting mistakes.
We sincerely appreciate the time you've spent on manuscript revision.
Round 2
Reviewer 2 Report
The manuscript "Effects of high voltage electrical discharge (HVED) on endogenous hormone and polyphenol profile in wheat" by Tihana Marček, Kamirán Áron Hamow, Tibor Janda, Eva Darko presents a study on metabolic changes indicated a stress-related response that can contribute germination in wheat with a special perspective on the synthesis of bioactive gibberellins.
The authors answered most of the questions asked in the review. I still think that the work is difficult to understand.
In addition, I recommend that the authors redo the conclusion, since it simply repeats the results, but does not contain conclusions. Perhaps the last statement is of value, but it is better to demonstrate it in the form of a diagram again in the discussion, rather than in the conclusions.
If this is done, the article may be published.
